# Identification and Characterization of MiRNAs in *Coccomyxa subellipsoidea* C-169

**DOI:** 10.3390/ijms20143448

**Published:** 2019-07-13

**Authors:** Runqing Yang, Gu Chen, Huifeng Peng, Dong Wei

**Affiliations:** School of Food Science and Engineering, South China University of Technology, Guangzhou 510640, China

**Keywords:** *Coccomyxa subellipsoidea* C-169, miRNA, lipid metabolism, transcription/translation factor, CO_2_ supplementation

## Abstract

*Coccomyxa subellipsoidea* C-169 (C-169) is an oleaginous microalga which is promising for renewable biofuel production. MicroRNAs (miRNAs), as the pivotal modulators of gene expression at post-transcriptional level, are prospective candidates for bioengineering practice. However, so far, no miRNA in C-169 has been reported and its potential impact upon CO_2_ supplementation remains unclear. High-throughput sequencing of small RNAs from C-169 cultured in air or 2% CO_2_ revealed 124 miRNAs in total, including 118 conserved miRNAs and six novel ones. In total, 384 genes were predicted as their potential target genes, 320 for conserved miRNAs and 64 for novel miRNAs. The annotated target genes were significantly enriched in six KEGG pathways, including pantothenate and CoA biosynthesis, C5-branched dibasic acid metabolism, 2-oxocarboxylic acid metabolism, butanoate metabolism, valine, leucine and isoleucine biosynthesis and alpha-linolenic acid metabolism. The miRNAs’ target genes were enriched in lipid metabolism as well as RNA-interacting proteins involved in translation, transcription and rRNA processing. The pioneering identification of C-169 miRNAs and analysis of their putative target genes lay the foundation for further miRNA research in eukaryotic algae and will contribute to the development of C-169 as an oleaginous microalga through bioengineering in the future.

## 1. Introduction

Human activities such as continuous use of fossil fuels, deforestation and intensive industrialization have led to rising production of greenhouse gases [1]. Oil-rich microalgae can grow rapidly through photosynthesis, assimilating CO_2_ and accumulating lipids simultaneously [2,3]. They have relatively high productivity, high lipid yields, and do not compete with crops or forestry for arable land and clean water [4]. As well, algae oil is a good raw material for the preparation of biofuels, thus making the oil-rich microalgae a good candidate for the production of renewable clean biofuels, saving energy and reducing emissions [5,6,7]. *Coccomyxa subellipsoidea* C-169 (hereafter C-169) is non-kinetic single-cell green algae found in Polar Regions. It is resistant to low-temperatures and high light and is the first polar genome to be sequenced [8]. The cell walls of C-169 are thin and brittle, and thus conducive to cell disruption and oil extraction [8]. Our previous studies indicated that supplementation of 2% CO_2_ increases the overall biomass productivity and fatty acid content of C-169 to 222 mg L^−1^ day^−1^ and 48.5%, respectively [9]. Transcriptomic analysis of C-169 with 2% against 0.04% CO_2_ unveiled that it employed a global and collaborative regulation on gene expression to assimilate more carbon and maintain the carbon/nitrogen balance, thus providing sufficient metabolic energy and an abundant carbon skeleton to sustain rapid growth and lipid accumulation upon elevated CO_2_ [9]. However, how the global and collaborative regulation of gene expression was achieved remains elusive.

MicroRNAs (miRNAs) are endogenous, non-protein coding RNAs with a single chain, which could post-transcriptionally regulate the expression of messenger RNAs in plants, animals and some viruses [10,11,12,13,14]. Mature miRNAs (20–25 nucleotides long) are produced from longer pri-RNA by nuclease cleavage processes [15]. They recognize target mRNAs by complementary base pairing and assemble into RNA-induced silencing complexes to degrade target mRNA or inhibit mRNA translation [16]. Therefore, miRNAs can regulate various essential biological processes such as development, growth, environmental adaption, stem cell division and apoptosis [17,18,19,20].

Compared with the large amount of miRNAs reported in various plants and animals [10,11,21,22], only limited reports of miRNAs were available in algae. To date, comprehensive miRNA profiles mainly revealed by high throughput sequencing were reported in diatom *Phaeodactylum tricornutum* [23], four green algae *Chlamydomonas reinhardtii* [24,25], *Volvox carteri* [26,27], *Botryococcus braunii* [28], *Dunaliella salina* [29] and four red algae *Chondrus crispus* [30], *Eucheuma denticulatum* [31], *Porphyridium purpureum* [32], *Porphyridium cruentum* [33], as well as two brown algae *Ectocarpus siliculosus* [34] and *Saccharina japonica* [35]. Recent studies in *Chlamydomonas reinhardti* indicated that induced expression of endogenous microRNA (miRNA1166.1) could enhance H_2_ production [36], while artificial miRNAs were manipulated successfully to improve the production of H_2_ and fatty acids [28,37,38]. Detailed analysis revealed that miRNAs in *Chlamydomonas* could regulate gene expression and metabolism by destabilizing and inhibiting the translation of coding regions in mRNA [39]. Taken together, it is suggested that miRNA may also have essential regulatory roles in unicellular microalgae [40]. Thus, exploration of endogenous miRNA in C-169 is of great significance to further apply it as a biofuel or bioproduct resource. However, to our knowledge, so far there is no report on C-169 miRNAs, and their potential function remains elusive.

In this study, we identified and characterized miRNAs in C-169, and analyzed the corresponding target genes and their potential regulation upon CO_2_ supplementation. Detailed analysis of endogenous miRNAs and their regulation on target genes would provide a foundation for further bioengineering practice, and contribute to the development of C-169 as an oleaginous microalga through bioengineering in the future.

## 2. Results and Discussion

### 2.1. High-Throughput Sequencing of Small RNAs in C-169

Small RNA libraries were prepared from C-169 cultured under air (0.04% CO_2_, termed AG) and 2% CO_2_ supplementation (termed CG) with three replicates. These small RNA libraries were subjected to deep sequencing and, in total, 8409893 and 8102757 raw reads were attained for the AG and CG groups, respectively. Following the bioinformatics pipeline shown in Appendix A, after removing junk reads and filtering out invalid reads, 5142305 and 4856938 valid reads of length between 18 and 25 nt remained for the AG and CG groups, respectively (Table 1). Their size distribution indicated that the most frequent size length was 20 nt in AG (23.70%) and 22 nt in CG (24.67%) (Figure 1a). More than half of these reads were mapped to the C-169 genome in the AG (56.65%) and CG (66.94%) groups (Table 1). These valid reads were collected for further identification and prediction of miRNA in C-169. The unique reads in AG and CG group were 137160 and 228512, respectively.

### 2.2. Identification of Mirnas in C-169

The resulting unique reads were annotated against conserved miRNAs of other species in miRbase 21.0 or predicted using the criteria described in the methods. A total of 124 miRNAs were identified in C-169, including 118 conserved and six novel miRNAs that were named from csl-nmiRNA1 to csl-nmiRNA6. The novel miRNA precursor structures are shown in Figure 1c. The length distribution of the identified miRNAs ranged from 18–24 nt, peaking at 21 nt (57.52%) and followed by 22 nt (Figure 1b). The identified conserved miRNAs were grouped into 73 miRNA families according to miRase 21.0 (Table 2). The largest miRNA family was miR166, which had 11 members, while a majority (79.45%) of detected families had only one member (Figure 2). Similarity and conservation were observed among the miRNAs identified here in C-169 and the reported miRNAs families in microalgae *Chondrus crispus* [30], *Porphyridium purpureum* [32] and *Eucheuma denticulatum* [31]. At least nine miRNA families, including 157, 164, 166, 167, 168, 397, 408, 482, 535, existed in all four algae mentioned here (Figure 2) and in *Botryococcus braunii* [28]. Members of these 73 miRNA families identified in C-169 were also distributed in 25 plant species (Appendix A). For example, 26 and 25 families also had members in *Oryza sativa* and *Glycine max*, respectively, thus suggesting the conservation of miRNAs in C-169, as well as their regulatory functions might be conserved among different species. Details of the conserved and novel miRNAs identified in C-169 are listed in Appendix A, respectively.

### 2.3. Validation of miRNAs in C-169 Through qRT-PCR

To validate the credibility of high-throughput sequencing data, RNAs were extracted from freshly cultured C-169 under AG and CG conditions. Eight miRNAs, including six conserved miRNAs and two novel ones, were subjected to quantitative reverse transcription PCR validation. The co-relationship R^2^ value between the qRT-PCR and sequencing data was 0.8258, confirming the credibility of the identified miRNAs in C-169 (Appendix A).

### 2.4. Functional Analysis of the Putative Target Genes of C-169 miRNAs

The miRNAs usually bind to the target mRNA and regulate gene expression by cleaving the target mRNA or inhibiting target mRNA translation. Using TargetFinder (https://github.com/carringtonlab/TargetFinder) and the criteria described in methods, 320 potential target genes for the 118 conserved miRNAs, and 64 targets for the six novel miRNAs were predicted (Appendix A). In total, 33 of these genes were predicted as the target of more than one miRNA.

In order to determine the potential function of the miRNA target genes, BLASTX querying against the protein database, gene ontology (GO), and KEGG pathway analysis were used to characterize these potential target genes. In total, 230 target genes of 49 miRNAs were associated with 768 GO terms according to the gene ontology consortium (Appendix A). They were significantly enriched in 12 terms (*p* < 0.01) (Figure 3a), such as GO:0000166 and GO:0006357. GO:0000166, which are involved with nucleotide binding in the molecular function and had 13 associated target genes. GO:0006357, which was involved in the biological process and involved in regulation of transcription by RNA polymerase II had four associated target genes. KEGG pathway analysis indicated that 137 target genes were involved in 87 pathways (Appendix A). The top 20 enriched pathways were identified with 51 genes, in which, 2-oxocarboxylic acid metabolism, ko01210; pantothenate and CoA biosynthesis, ko00770; C5-Branched dibasic acid metabolism, ko00660; butanoate metabolism, ko00650; valine, leucine and isoleucine biosynthesis, ko00290 and alpha-linolenic acid metabolism, ko00592 were significantly enriched (*p* < 0.05) (Figure 3b). These six enriched pathways could be connected through assembly protein and metabolic enzymes, such as clathrin assembly protein (58860), lipase (57158), acetolactate synthase (19013) and acyl-coenzyme A oxidase 2 (54075), as the network proposes in Figure 4.

### 2.5. Functional Analysis of the Putative Target Genes in Metabolism and Transcription and Translation Process

Among the 215 annotated target genes, 80 were annotated as metabolic enzymes or important components involved in C-169 metabolism, such as Cytochrome b-c1 complex subunit 9 (65894) of oxidative phosphorylation, Photosystem I reaction center subunit VI (53060) of photosynthesis, glutamate decarboxylase 5 (38146) of amino acids biosynthesis, and aconitate hydratase (27212) of the TCA cycle (Appendix A). Interestingly, 21 annotated target genes were enriched in lipid metabolism (Appendix A, Figure 5). The expression ratio upon CO_2_ supplementation (CG/AG) of miRNAs and their putative target genes were compared using the published transcriptomic data [9], and the opposite pattern was found in some pairs of miRNA and their target genes involved in lipid metabolism. For example, 54075 and 54823 encode long-chain acyl-CoA oxidase and fatty acid beta-oxidation multifunctional protein MFP2, both of which were involved in the oxidation and degradation of fatty acid. These two genes were significantly repressed in C-169 upon CO_2_ supplementation and contributed to the accumulation of lipids [9]. Impressively, consistent upregulation was observed here in their corresponding miRNAs, miR7700 and miR1511 (Appendix A). Other target genes involved in lipid catabolism included 25019 (12-oxophytodienoic acid reductase) in alpha-linolenic acid degradation, 47839 (Glycerol-3-phosphate dehydrogenase) and 61026 (glycerophosphodiester phosphodiesterase) in glycerophospholipid catabolism, and five lipases, as 39910, 39938, 47533, 57158 and 63589 to hydrolyze lipid into di- and monoglycerides, glycerol, and free fatty acids. Meanwhile, lipid biosynthesis genes were also targeted by miRNAs, such as 49000 (Fatty acid synthase)-the critical enzyme of fatty acid biosynthesis, 49349 (Digalactosyldiacylglycerol synthase 1) and 64764 (Diacylglycerol O-acyltransferase 2A)-the enzymes essential for glycerolipid synthesis, 43553 (Beta-galactosidase 17) for glycosphingolipid biosynthesis, as well as 48005 (very-long-chain 3-oxoacyl-CoA reductase) for the elongation of long chain fatty acid and three genes (41238, 44340, 52322) encoding desaturases for generating unsaturated fatty acid.

Opposite regulation patterns were also observed in transcripts of miR5072, miR167e, miR9558-5p, miR156a and their predicted target 39545, 19013, 64764 and 68051 (Figure 6 and Appendix A). However, the opposite expression pattern was not obvious in the other pairs of miRNA and their putative targets. Similar results were reported in *Chlamydomonas reinhardtii* as well. Though miRNAs were found differentially expressed under different tropic conditions, the majority of their predicted targets didn’t show opposite changes in transcript abundance [25]. Actually, a comprehensive global analysis of protein synthesis and RNA abundance influenced by miRNAs revealed that endogenous miRNAs in *Chlamydomonas* could regulate gene expression both by destabilization of the mRNA through cleavage and by translational repression [39]. Further research is needed to verify the detailed mode of action for miRNAs in C-169.

According to the published data in other microalgae, miRNA target genes in *Eucheuma denticulatum* were highlighted to enrich in the porphyrin and chlorophyll metabolic pathway [31]. Peroxisome proliferator-activated receptor (PPAR) signaling pathways enriched in miRNA target genes were reported in *Chondrus crispus* [30], while phagocytosis and proteasome were emphasized as the most significant enriched pathway for conserved and novel miRNAs respectively in *Porphyridium purpureum* [32]. Lipid metabolism has not been mentioned in miRNA targets enriched pathways in microalgae other than C-169. Here, among the 80 annotated miRNA target genes related to metabolism, more than one fourth were involved in lipid metabolism of C-169. This suggests the important role of miRNAs in regulating lipid metabolism of this oleaginous microalga, thus rendering miRNAs bioengineering practices promising for renewable biofuel or bioproduct production in C-169. 

The interplay between miRNAs and transcription factors has been emphasized in plants, and many miRNA targets have been revealed as transcription factors (TFs), which play important roles in regulating the plant growth, development and environmental response [41,42,43,44,45]. Here, we found in C-169, among the 215 annotated miRNAs target genes, 49 which were related to transcription or translation (Appendix A). Firstly, target genes included several helicases (16341, 30872, 65316, 65482) involved in RNA splicing in spliceosome and 52093 (U4/U6 small nuclear ribonucleoprotein Prp3) as one of the spliceosome components. Splicing related target genes also included pre-mRNA splicing factors 44563, 45296 and 48307, splicing coactivator 61159, mRNA 3′ end processing factor 68114 (cleavage and polyadenylation specificity factor subunit) and tRNA-splicing ligase RtcB homolog 46227. Splicing is a critical step in RNA biogenesis, but whether these splicing related components are involved in miRNA processing needs further investigation. Secondly, transcription related target genes included RNA polymerase subunit 21078, RNA polymerase II transcription initiation factor TFIIH subunit 4 18513, transcription elongation factor SPT6 47678 and transcription termination factor 47507 and 48741 (P-loop containing nucleoside triphosphate hydrolase protein). In the meantime, several transcription factors were targeted, such as RNA polymerase III basal transcription factor C subunit 64071, transcription factors 65593 (homeobox protein cut-like transcription factor), 67545 (pheromone receptor transcription factor), 66962 (bZIP containing protein), 57853 (transcription factor containing Myb domain), as well as 66493 and 59006 (AP2-domain-containing protein). Recently, the important regulatory roles of transcription factors have been confirmed in microalga. For example, in *Phaeodactylum tricornutum*, transcription factor bZIP14 was identified as a conserved regulator in the TCA cycle during nitrogen starvation [46]. In *Tisochrysis lutea*, transcription factors MYB-2R_14 and NF-YB_2 TFs were found related to photosynthesis, oxidative stress response and triacylglycerol synthesis, while GATA_2, MYB-rel_11 and MYB-2R_20 were likely to be related to nitrogen uptake or carbon and nitrogen recycling [47]. Overexpression or repression of specific transcription factors could be an efficient strategy in the bioengineering of microalgae. For instance, overexpression of a bZIP transcription factor enhanced biomass and lipid productivity in *Nannochloropsis salina* [48]. Here, several transcriptional regulators were revealed as the miRNA target, rendering it possible to manipulate transcription factors through overexpression of miRNA or miRNA mimicry. Thirdly, miRNA target genes were enriched in translation related process. Significant enrichment was found in translation initiation factors. Impressively, six out of forty two translation initiation factors of C-169 were predicted as the miRNA target genes, as 21475, 25465, 53345, 56024, 64334, and 65695. As well, translation elongation associated factor 18056 (Diphthine methyl ester synthase) and translation factor GUF1 47382 were targeted. MiRNAs were also predicted to target on several ribosome biosynthesis proteins, such as small subunit ribosomal protein S5 57673, U3 small nucleolar RNA-associated protein 22 65874, ribosome maturation factor RimM 54037, ribosome-binding factor A 41503, RNA 3′-terminal phosphate cyclase 33699, and ARM repeat-containing protein 13771, which were involved in mRNA localization in ribosome.

Analogously, global analysis of the mode of action of *Chlamydomonas* miRNAs indicated that mRNAs targets are subject to either strong translational repression or RNA destabilization is enriched with those encoding RNA-interacting proteins involved in translation, transcription and rRNA processing [39]. Whether it is common to other green alga needs further investigation. To identify functional miRNA pathways in C-169, we also searched the database for the presence of the potential factors involved in RNA silencing pathways and those which contain a conserved motif. Two PIWI-domain-containing proteins (encoded by 56024 and 56023), typical for argonaute, were found in C-169. Interestingly, 56024 were annotated as translation initiation factors and predicted to be the target of miRNA. As well, two DEAD-domain-containing proteins (encoded by 34952 and 63003) were revealed as ATP-dependent RNA helicase and involved in RNA-induced silencing complex (RISC), suggesting that miRNA pathways could be functional in C-169. The detailed roles and mechanism of these components in miRNA generation and action could be further investigated.

MicroRNAs (miRNAs), thus are regarded as promising candidates in bioengineering modification [49]. Overexpression of miRNA or expression of miRNA target mimicry are potential strategies to enhance or repress microRNA activity, respectively. For instance, in green microalga *Chlamydomonas reinhardtii,* the production of fatty acid was increased by inhibiting phosphoenolpyruvate carboxylase [50]. Target mimicry transcripts could specifically trap members of a miRNA family, thus block their activity on endogenous targets [51,52]. For example, in legume *Medicago truncatula*, miR396 inactivation by target mimicry increased target gene expression, leading to higher biomass of root and more efficient infection by symbiotic fungi *arbuscular mycorrhizal* [53]. Thus, miRNA target mimicry expression could result in the superactivation of endogenous targets and the downstream process. Here in C-169, miRNAs were identified and predicted to target on metabolic enzymes as well as transcription and translation related genes, providing a potential strategy to manipulate regulation through overexpression of miRNA or miRNA target mimicry. However, since data here was collected only from one time point upon CO_2_ supplementation and further research is needed to verify the detailed mode of action for miRNAs in C-169. Such further investigation of the miRNA and their target genes in C-169, especially under different conditions, might provide useful tools to develop C-169 as an oleaginous microalga through miRNA bioengineering in the future.

## 3. Materials and Methods

### 3.1. Algal Strain and Culture Conditions

*Coccomyxa subellipsoidea* C-169, strain number NIES 2166, was obtained from the Microbial Culture Collection of National Institute for Environmental Studies in Japan. C-169 was inoculated into a 250 mL Erlenmeyer flask containing 100 mL Bold’s Basal Medium (BBM) for autotrophic culture to the exponential phase (OD680 = 0.8). Subsequently, the algal was inoculated with a 10% inoculum size into a 150 mL Erlenmeyer flask containing 60 mL BBM medium and cultured in 0.04% and 2% CO_2_ carbon dioxide incubator, respectively. The illumination intensity was set at 60 µmol·m^–2^·s^−1^ and the temperature was 26 °C. On the fourth day, cells were collected through centrifugation, washed by Tris-EDTA (TE) buffer and frozen for 5 min in liquid nitrogen, and then stored at −80 °C for RNA extraction and miRNA analysis.

### 3.2. RNA Extraction, Library Construction and Sequencing

Total RNA was extracted using Trizol (Invitrogen, CA, USA) following the manufacturer’s procedure. The total RNA quantity and purity were analyzed through Bioanalyzer 2100 (Agilent, CA, USA). Approximately 1 ug of total RNA was used to prepare a small RNA library according to the protocol of TruSeq Small RNA Sample Prep Kits (Illumina, San Diego, USA). Then single-end sequencing was performed on Illumina Hiseq2500 at the LC-BIO (Hangzhou, China).

### 3.3. Sequencing Data Analysis

Briefly, the raw reads were subjected to ACGT101-miR (LC Sciences, Houston, Texas, USA), an in-house program to remove adapter dimers, junk, low complexity, common RNA families such as rRNA, tRNA, snRNA, snoRNA and repeats. Then, in order to identify known miRNAs, novel 3p- and 5p- derived miRNAs, unique sequences (18–25 nt) were mapped to miRNA precursors of different plant species in miRbase 21.0 (http://mirbase.org) by BLAST search. During alignment, length change at both 5′ and 3′ ends and one mismatch in the sequence were allowed. The unique sequences were identified as known miRNAs when mapping to specific species mature miRNAs in hairpin arms. The remaining sequences were mapped to other precursors in miRbase 21.0 (http://mirbase.org) by BLAST search to identify conserved miRNAs, and the precursors of conserved miRNAs were blasted against the C-169 genomes to confirm the locations. To identify novel miRNAs, the unmapped sequences were blasted against the C-169 genomes, and the sequences existed in the structures of hairpin RNA were predicated from the flanking 120 nt sequences using RNA fold software (http://rna.tbi.univie.ac.at/cgi-bin/RNAfold.cgi). The following criteria were used: (1) number of base pairs in the stem region of the predicted hairpin ≥16; (2) number of nucleotides in one bulge in stem ≤12; (3) minimal folding free energy ≤−15 kCal/mol; (4) length of hairpin loop ≤200; (5) length of hairpin, up and down stems as well as terminal loop ≥50; (6) number of biased errors in one bulge in mature region ≤2; (7) number of nucleotides in one bulge in mature region ≤4; (8) number of base pairs in the mature region of the predicted hairpin ≥12; (9) number of errors in mature region ≤ 4; (10) number of biased bulges in mature region ≤2; (11) percent of mature in stem ≥80. The miRNAs were classified into families according to miRase 21.0.

### 3.4. Target Gene Prediction and Function Analysis

TargetFinder [54] (https://github.com/carringtonlab/TargetFinder) was used to predict putative target genes, following the principle of penalty system based on complementary base pairing: (1) mismatches: penalty 1; (2) G: U pairing: penalty 0.5; (3) if both of the above conditions occur on sRNA sequences at positions 2–13 of the 5′end, the penalty is doubled; (4) the default of four points or less is a potential target site. Functional classification of potential target genes of miRNA was conducted according to annotation in gene ontology [55] (GO) (http://geneotology.org/) and the pathway analysis was carried out according to KEGG [56] (http://www.genome.jp/kegg).

### 3.5. Quantitative RT-PCR

Quantitative RT-PCR was performed on CFX96 Touch™ Deep Well Real-Time PCR Detection System (BIO RAD, Hercules, California) using two-step kit, Mir-XTM miRNA First- Strand Synthesis Kit and SYBR Premix Ex Taq II kit (TaKaRa Biotech Co., Dalian, China). The first step was to convert miRNA to cDNA by poly(A) polymerase and reverse transcriptase which were included in the mRQ Enzyme Mix. Gene U4 was used as an internal control according to reference [57]. The 3′ primer used was the mRQ 3′ Primer supplied in the kit and the specific 5′ primers for each miRNA were listed in Appendix A. Each qRT-PCR reaction containing 2 μL cDNA, 1 μL 3′ and 5′ primer, 12.5 μL SYBR Premix and ddH2O was added to make a final volume of 25 μL. Amplification program was set as 95 °C 30 s, 40 cycles at 95 °Cfor 5 s and 60 °C for 30 s followed by disassociation stage as instructed by user’s manual. Experiments were performed in triplicate. The relative number of gene transcripts was normalized to that of reference gene U4.

## 4. Conclusions

For the first time, 124 miRNAs including 118 conserved miRNAs and six novel miRNAs were identified in C-169. Totally 384 target genes were predicted, 320 for conserved and 64 for novel miRNAs. The annotated target genes were significantly enriched in KEGG pathways, such as pantothenate and CoA biosynthesis, C5-branched dibasic acid metabolism, 2-oxocarboxylic acid metabolism, butanoate metabolism, valine, leucine and isoleucine biosynthesis and alpha-linolenic acid metabolism. The miRNAs target genes were enriched in lipid metabolism over basal metabolism. The predicted miRNA targets were significantly enriched in genes encoding RNA-interacting proteins involved in translation, transcription and rRNA processing, such as transcription factors and translation initiation factors. As the first study to identify miRNA in C-169, it lays the foundation for further miRNA research in eukaryotic algae, as well as contributing to the development of C-169 as an oleaginous microalga through bioengineering in the future.

## Figures and Tables

**Figure 1 ijms-20-03448-f001:**
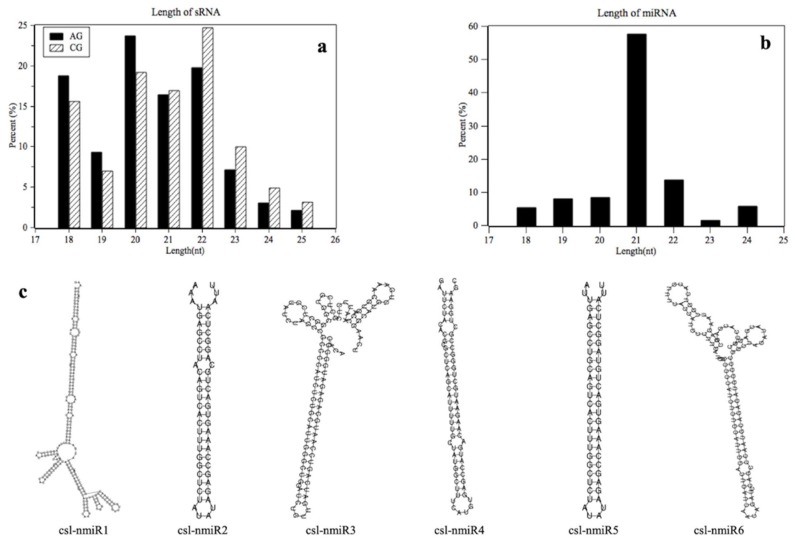
(**a**) Length distribution of sRNA. Length distribution of valid reads after removing t/r/sn/sno RNAs in the small-RNA library. (**b**) Length distribution of identified MicroRNAs (miRNAs). The length distribution of 124 miRNAs, including 118 conserved and six novel ones. (**c**) The precursor structures of six novel miRNAs.

**Figure 2 ijms-20-03448-f002:**
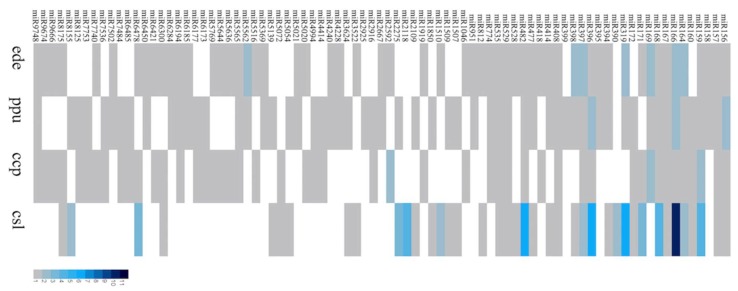
Comparison of conserved miRNA families between C-169 and another alga. The conserved miRNA identified so far in *Chondrus crispus* (ccp), *Porphyridium purpureum* (ppu), *Eucheuma denticulatum* (ede) and C-169 (csl) are listed for comparison. Color-coding is used to indicate the number of miRNA members in each family.

**Figure 3 ijms-20-03448-f003:**
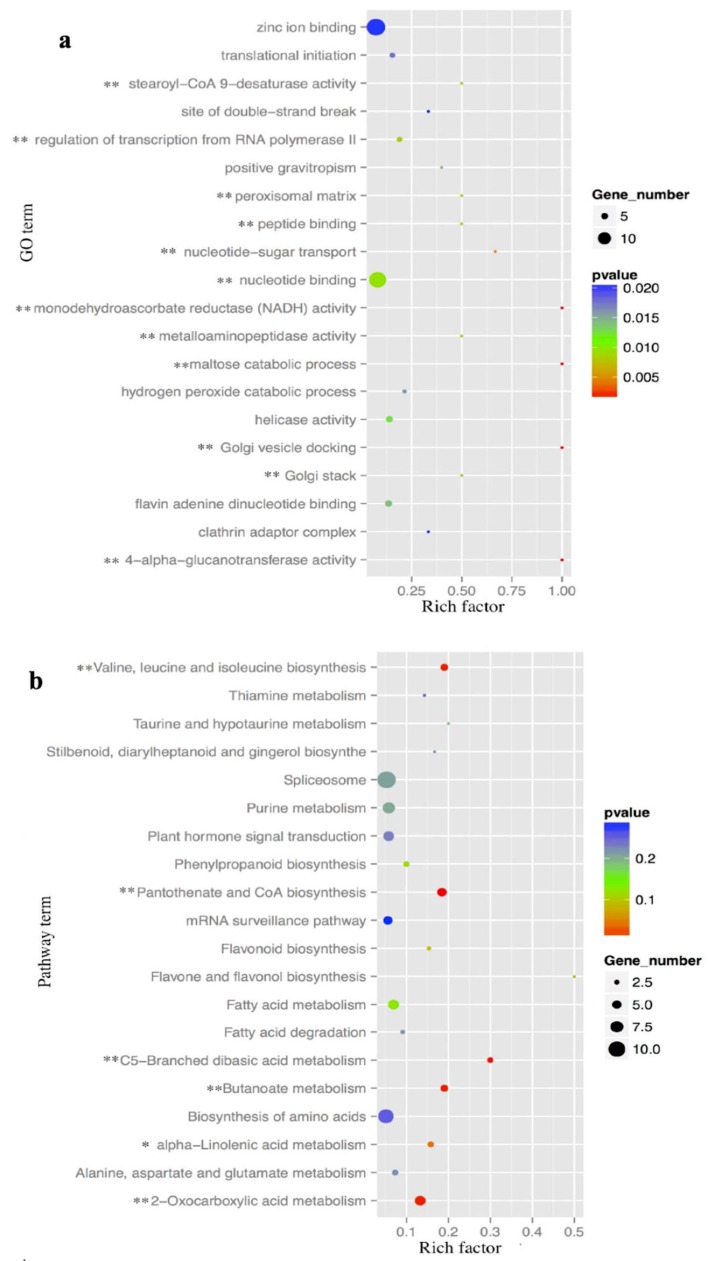
The enrichment of putative miRNA target genes in gene ontology (GO) and KEGG terms. (**a**) Twenty miRNA target genes mostly enriched GO terms. (**b**) Twenty miRNA target genes mostly enriched KEGG terms. The size of the circle is proportional to the numbers of target genes that are involved. The color of the circle indicates the significance of enrichment. The symbols ** and * indicate *p* < 0.01 and *p* < 0.05, respectively. The *x*-axis represents the rich factor.

**Figure 4 ijms-20-03448-f004:**
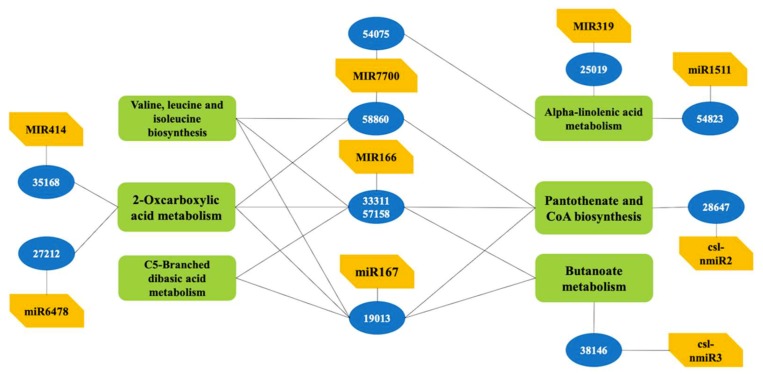
Network among six enriched KEGG pathways. These six pathways are C5-branched dibasic acid metabolism, pantothenate and CoA biosynthesis, 2-oxocarboxylic acid metabolism, butanoate metabolism, valine, leucine and isoleucine biosynthesis and Alpha-Linolenic acid metabolism.

**Figure 5 ijms-20-03448-f005:**
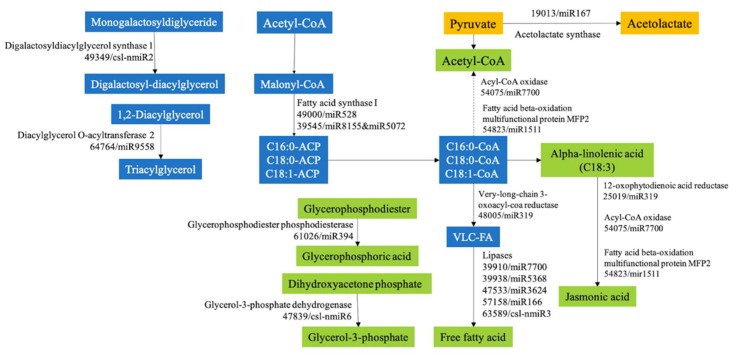
The target genes that participated in lipid metabolism.

**Figure 6 ijms-20-03448-f006:**
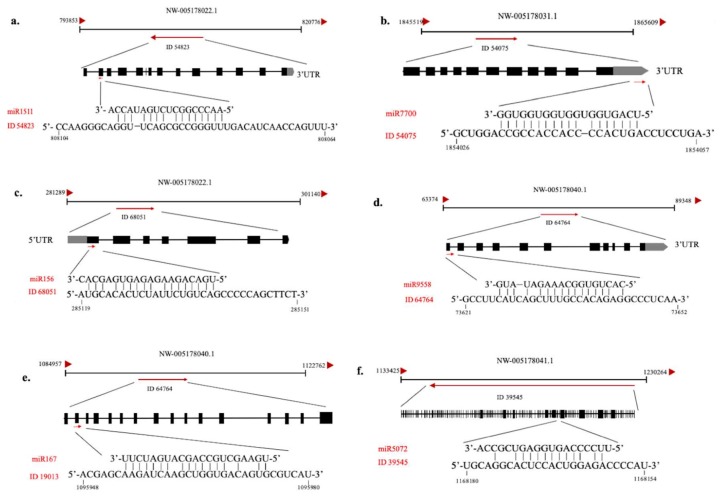
Analysis of the miRNAs and their predicted target genes including miR1511/54823 (**a**), miR7700/54075 (**b**), miR156/68051 (**c**), miR9558/94764 (**d**), miR167/19013 (**e**), miR5072/39545 (**f**), respectively.

**Table 1 ijms-20-03448-t001:** Summary of small RNA sequencing data in the cultured under air (AG) and 2% CO_2_ supplementation (CG) library of *Coccomyxa subellipsoidea* C-169 (C-169).

Read	AG	CG
Raw data	8409893	8102757
3′ adaptor sequence & length filter	2627653	2254543
Junk reads	5554	7345
Valid reads	5142305	4856938
Genome mapped	2913297	3251099
Genome mapped (%)	56.65%	66.94%
Unique reads	137160	228512
miRNA mapped	24974	27025
miRNA mapped (%)	0.49%	0.55%

**Table 2 ijms-20-03448-t002:** Summary of C-169 conserved miRNA families and novel miRNAs.

Family	No of Members	miRNA Name	miRNA Sequence (5′–3′)
Conserved miRNA	133		
miR156	1	csl-miR156a	TGACAGAAGAGAGTGAGCAC
miR157	1	csl-miR157a-5p	TTGACAGAAGATAGAGAGCAC
miR159	4	csl-miR159a	TTTGGATTGAAGGGAGCTCTA
		csl-miR159a-5p	AGCTCCCTTCGATCCAATC
		csl-miR159a-3p	CTTGGATTGAAGGTAGCTCT
		csl-miR159b	TTTGGATTGAAGGGAGCTCTATT
miR160	1	csl-miR160a-3p	GCGTATGAGGAGCGAAGCATA
miR162	1	csl-miR162a-3p	ATCGATAAACCTCTGCATCCAG
miR164	2	csl-miR164a	TGGAGAAGCAGGGCACGTGCA
		csl-miR164a-1	TGTAGAAGCAGGGCACATGCC
miR166	11	csl-miR166a-3p	TCGGACCAGGCTTCATTCCCC
		csl-miR166a-5p-1	GGAATGTTGTCTGGTTCAAGG
		csl-miR166a-5p	GGAATGTTGTCTGGCTCGGGG
		csl-MIR166b-3p	GATAATGATAATGATAATG
		csl-MIR166b-5p	TAATGATAATGATAATGATAAT
		csl-miR166e-3p	TGGAACCAGGCTTCATTCCCC
		csl-miR166h-5p	GGAATGTTGGCTGGCTCGAGG
		csl-miR166i-3p	TCTCGGATCAGGCTTCATTCC
		csl-miR166j	CCGGACAAGGCTTCATTCCCC
		csl-miR166k-5p	GGTTTGTTGTCTGGCTCGAGG
		csl-miR166m	GCTGACCAGGCTTCATTCCCC
miR167	1	csl-miR167e	TGAAGCTGCCAGCATGATCTT
miR168	4	csl-miR168	TCTCTTGGTGCAGGTCGGGAA
		csl-miR168a-3p-1	CCCGCCTTGCATCAACTGAAT
		csl-miR168a-3p	CCCGCCTTGCACCAAGTGAAT
		csl-miR168a-5p	TCGCTTGGTGCAGGTCGGGAA
miR171	3	csl-miR171a	TTGAGCCGCGTCAATATCTCC
		csl-miR171h	TGAGCCGAACCAATATCACTC
		csl-miR171k-3p	TTGAGCCGCGCCAATATCACT
miR172	1	csl-miR172a	AGAATCTTGATGATGCTGCAT
miR319	6	csl-miR319	GTGGGACTGAAGGGAGCTCCC
		csl-MIR319a-5p	TAACACGTCCGGGTTGCT
		csl-miR319a-3p	TTGGACTGAAGGGTGCTCCCT
		csl-miR319a-5p	AGAGCTTCCTTCAGTCCACTCA
		csl-miR319b	GCTTGGACTGAAGGGAGCTCCTTC
		csl-miR319b-5p	AGAGCGTCCTTCAGTCCACTC
miR390	2	csl-miR390-3p	CGCTATCTATCCTGAGCTCC
		csl-miR390-5p	AAGCTCAGGAGGGATAGCGCC
miR393	1	csl-miR393a	TCCAAAGGGATCGCATTGATT
miR394	1	csl-miR394a-5p	TTGGCATTCTGTCCACCTCC
miR396	6	csl-miR396a-3p-1	GTTCAATAAAGTTGTGGGAAG
		csl-miR396a-3p	CTCAAGAAAGCTGTGGGAGA
		csl-MIR396a-5p	TTGATGATCTTCTTCAAAA
		csl-miR396a-5p-1	TTCCACAGCTTTCTTGAACTG
		csl-miR396a-5p	TCCACAGGCTTTCTTGAACTG
		csl-miR396b	TTCCACAGCTTTCTTGAACTTTT
miR397	2	csl-miR397a	TCATTGAGTGCAGCGTTGATG
		csl-miR397b-5p	TTGAGTGCAGCGTTGATGAACC
miR398	1	csl-miR398c	CATGTGTTCTCAGGTCGCCCC
miR403	1	csl-miR403-3p	TTAGATTCACGCACAAACTCG
miR408	1	csl-miR408-3p	TGCACTGCCTCTTCCCTGGCT
miR414	1	csl-MIR414-5p	ACATCATCATCGTCATCATC
miR444	1	csl-miR444c	TGCAGTTGTTGTCTCAAGCTT
miR477	1	csl-miR477a	ACTCTCCCTCAAGGGCTTCTGA
miR482	6	csl-miR482-3p	TTCCCAATTCCGCCCATTCCTA
		csl-miR482-5p	GGAATGGGCTGATTGGGAAGC
		csl-miR482a-3p	TTCCCAAGCCCGCCCATTCCAA
		csl-miR482a-5p	GGAATGGGCTGTTTGGGATG
		csl-miR482b-3p	TCTTCCCTACACCTCCCATACC
		csl-miR482c	TCTTTCCTACTCCACCCATTCC
miR528	1	csl-miR528-5p	TGGAAGGGGCATGCAGAGGAG
miR529	1	csl-miR529-5p	AGAAGAGAGAGAGTACAGCCT
miR535	1	csl-miR535-5p	TGACAACGAGAGAGAGCACGC
miR812	1	csl-miR812a	GACGGACGGTCAAACGTTGGACAC
miR827	1	csl-miR827	TTAGATGACCATCAGCAAACA
miR894	1	csl-miR894	CGTTTCACGTCGGGTTCACCA
miR946	1	csl-miR946	CAGCCCTTCTCCTATCCACAAT
miR1314	1	csl-miR1314	CCGGCCTCGAATGTTAGGAGAA
miR1423	1	csl-miR1423-5p	GCAACTACACGTTGGGCGCTCGAT
miR1425	1	csl-miR1425-5p	TAGGATTCAATCCTTGCTGCT
miR1448	1	csl-miR1448	TCTTTCCAACGCCTCCCATACC
miR1507	1	csl-miR1507a	TCTCATTCCATACATCGTCTGA
miR1509	1	csl-miR1509a	TTAATCAAGGAAATCACGGTCG
miR1510	2	csl-miR1510b-3p	TGTTGTTTTACCTATTCCACCT
		csl-miR1510b-5p	AGGGATAGGTAAAACAACTAC
miR1511	1	csl-miR1511	AACCCGGCTCTGATACCA
miR1850	1	csl-miR1850.1	TGGAAAGTTGGGAGATTGGGG
miR1862	1	csl-miR1862e	CTAGATTTGTTTATTTTGGGACGG
miR1876	1	csl-miR1876	ATAAGTGGGTTTGTGGGCTGGCCC
miR2109	1	csl-miR2109-3p	GGAGGCGTAGATACTCACACC
miR2118	4	csl-miR2118a-3p	TTGCCGATTCCACCCATTCCTA
		csl-miR2118b	TTCCCAATGCCTCCCATTCCTA
		csl-miR2118f	TCCTGATGCCTCCCATTCCTA
		csl-miR2118o	CTCCTGATGCCTCCCAAGCCTA
miR2275	3	csl-miR2275b-3p	TTCAGTTTCCTCTAATATCTCG
		csl-miR2275b-5p	AGGATTAGAGGGAACTGAACC
		csl-miR2275c	AGAATTGGAGGAAAACAAACT
miR3522	1	csl-miR3522	TGAGACCAAATGAGCAGCTGA
miR3624	1	csl-miR3624-3p	TCAGGGCAGCAGCATACTA
miR3630	1	csl-miR3630-3p	TGGGAATCTCTCTGACGCT
miR4412	1	csl-miR4412-5p	TGTTGCGGGTATCTTTGCCTC
miR4413	1	csl-miR4413a	TAAGAGAATTGTAAGTCACTG
miR4995	1	csl-miR4995	GGCAGTGGCTTGGTTAAGGGAACC
miR4996	1	csl-miR4996	TAGAAGCTCCCCATGTTCTCA
miR5054	1	csl-miR5054	TCCCCACGGACGGCGCCA
miR5072	1	csl-miR5072	TTCCCCAGTGGAGTCGCCA
miR5077	1	csl-miR5077	GTTCACGTCGGGTTCACCA
miR5139	1	csl-miR5139	CGAAACCTGGCTCTGATACCA
miR5368	1	csl-MIR5368-3p	CTGGGATTGGCTTTGGGC
miR5372	1	csl-miR5372	TTGTTCGATAAAACTGTTGTG
miR5527	1	csl-miR5527	TCTCAGCCAGGGCAGTAACAG
miR5530	1	csl-miR5530	AGTGGTGTCGTATTACCTGCC
miR5724	1	csl-miR5724	AACCGCCGGTTCGATAAT
miR5770	1	csl-miR5770a	TTAGGACTATGGTTTGGACGA
miR5792	1	csl-miR5792	GATGACAGCGGTGGTTCGGACCTC
miR5796	1	csl-miR5796	TCATTCAGGATTGAAGCCGCC
miR5818	1	csl-miR5818	TCGAACTAGAAGGGCCAGGTT
miR6300	1	csl-miR6300	GTCGTTGTAGTATAGTGGT
miR6478	3	csl-miR6478-2	CCGACCTTAGCTCAGTTGGTAGA
		csl-miR6478-1	CCGGCCTTAGCTCAGTTGG
		csl-miR6478	CCGACTTTAGCTCAGTTGG
miR7700	1	csl-MIR7700-5p	TCAGTGGTGGTGGTGGTGG
miR8005	1	csl-MIR8005c-3p	TTTAGGGTTTAGGGTTTAGG
miR7972	1	csl-miR7972	TTGTCAGGCTTGTTATTCTCC
miR8155	2	csl-miR8155-1	GTAACCTGGCTCTGATACCA
		csl-miR8155	AACCTCGCTCTGATACCA
miR8175	1	csl-miR8175	TCGTTCCCCGGCAACGGCGCCA
miR9558	1	csl-MiR9558-5p	CACTGTGGCAAAGATATG
Novel	6		
csl-nmiR1	1	csl-nmiR1	AGGGCGTTCCGTCGGACGGGTT
csl-nmiR2	1	csl-nmiR2	GCCAAAGTGACTGCAGGCT
csl-nmiR3	1	csl-nmiR3	TCTCTGCATTGGGCTGGATCT
csl-nmiR4	1	csl-nmiR4	GCCATGACAAGAATGCTGGGCC
csl-nmiR5	1	csl-nmiR5	GCCTACAGTCACTTTGGCT
csl-nmiR6	1	csl-nmiR6	TGCAGAGATGGGACGGCT

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
