# Peer review of "Identification and Characterization of MiRNAs in *Coccomyxa subellipsoidea* C-169"

_ijms, 2019, doi:10.3390/ijms20143448_

Round 1

Reviewer 1 Report

The paper is focused on transcriptomic analysis of microRNAs in oleaginous trebouxiophyte alga Coccomyxa sp. C-169. The topic is timely, important and therefore suitable for the Physiology and Molecular Biology of Plants readership. The analyses are correctly performed. Though, some information was unclear or missing to me, so I have several following points/questions.

1) Figure 2 shows comparison of miRNA repertoire of C-169 with some red/brown algae. Why authors did not select also phylogenetically more closely related green algal species for such comparison?

2) I would appreciate comparison of total miRNA counts with also with other algae to have better idea about C-169 uniqueness or mediocrity in terms of the size of miRNA pool. In general, can the size of the miRNAs repertoire indicate something about the importance of this type of gene expression regulation? 

3) Generally, miRNAs can be complementary to sequences of genes or to intergenic regions. What was the case for C-169 species?

4) In my opinion, Discussion section should be better arranged and structured. Text accompanying Figure 5 and Figure 6 could be probably moved into Results together with respective figures.

5) Some miRNAs can serve as enhancers of the transcription. Was this potential funtion considered when performing the annotations?

6) The authors claim that their work is the first description of miRNA in oil producing alga, however, miRNA of another oleaginous alga, Botryococus, have been already published. Comparing miRNAs of those two species could help to identify miRNAs importatnt in oil production.

7) I spotted a few mistakes in English, eg.:

- line 196:  "but whether these splicing related

components involved in miRNA processing needs further investigation" - I think should be ...are involved...

- line 205: Tisochrysis lutea should be in italics

- line 307: The miRNAs target genes were enriched in lipid metabolism among basal metabolism - I think should be ...over basal...

Author Response

The paper is focused on transcriptomic analysis of microRNAs in oleaginous trebouxiophyte alga Coccomyxa sp. C-169. The topic is timely, important and therefore suitable for the Physiology and Molecular Biology of Plants readership. The analyses are correctly performed. Though, some information was unclear or missing to me, so I have several following points/questions.

1)      Figure 2 shows comparison of miRNA repertoire of C-169 with some red/brown algae. Why authors did not select also phylogenetically more closely related green algal species for such comparison?

Response 1: Figure 2 shows comparison of miRNA family and their member numbers among C-169 and some red/brown algae. Green algae were not included since their miRNA family member information was not available so far in the four species of green algae, Chlamydomonas reinhardtii, Volvox carteri, Botryococcus braunii and Dunaliella salina reported with miRNAs data. In Chlamydomonas reinhardtii and Volvox carteri, different expression of miRNAs was discussed under certain condition, lack of total miRNAs families’ information. In Botryococcus braunii, 42 known families were identified without members’ information in detail. In Dunaliella salina, the family information of miRNAs was not mentioned so far. Thus only the comparison of miRNA of C-169 with some red/brown algae was indicated in Figure 2. But a description of Botryococcus braunii miRNA families was added in revised manuscript.

2)      I would appreciate comparison of total miRNA counts with also with other algae to have better idea about C-169 uniqueness or mediocrity in terms of the size of miRNA pool. In general, can the size of the miRNAs repertoire indicate something about the importance of this type of gene expression regulation? 

Response 2: I think only when the data of size of the miRNAs repertoire are collected from more and more different time points and conditions, can the size indicate the importance of this type of gene expression regulation.

Since data so far were collected generally from one time point and identified under different culture condition in different algae, it is difficult to draw conclusion from such comparison.

3)      Generally, miRNAs can be complementary to sequences of genes or to intergenic regions. What was the case for C-169 species?

Response 3: MiRNAs identified here were complementary to the sequences of genes, including coding region and untranslated region (UTR) (Fig. 6b), but not the intergenic regions.

4)      In my opinion, Discussion section should be better arranged and structured. Text accompanying Figure 5 and Figure 6 could be probably moved into Results together with respective figures.

Response 4: Revised as suggested.

5)      Some miRNAs can serve as enhancers of the transcription. Was this potential function considered when performing the annotations?

Response 5: This potential function was considered when performing the annotations. But so far no solid evidence was attained thus it was not mentioned in the manuscript.

6)      The authors claim that their work is the first description of miRNA in oil producing alga, however, miRNA of another oleaginous alga, Botryococus, have been already published. Comparing miRNAs of those two species could help to identify miRNAs important in oil production.

Response 6: Revised as suggested, only mentioned this manuscript as the “the first description of miRNA in C169”, but not in oil producing alga.

Comparing miRNAs of those two species encounters obstacle that no detail function analysis was available in Botryococus so far. Further exploration and comparison will be performed in the future.

7) I spotted a few mistakes in English, eg.:

- line 196:  "but whether these splicing related components involved in miRNA processing needs further investigation" - I think should be ...are involved...

- line 205: Tisochrysis lutea should be in italics

- line 307: The miRNAs target genes were enriched in lipid metabolism among basal metabolism - I think should be ...over basal...

Response 7: They were revised as suggested.

Reviewer 2 Report

I've read with attention the paper entitled "Identification and characterization of miRNAs in 3 Coccomyxa subellipsoidea C-169" that is potentially of interest. The methodology applied is overall correct, the obtained results are reliable and adequately discussed. I only would suggest to change the final part of the introduction, when the authors write "Therefore, we identified and characterized miRNAs in C-169 through high-throughput sequencing of 63 small RNA fragments, analyzed their predicted target genes and discussed their potential regulation upon CO2 64 supplementation. " In fact, this is more a conclusion than a study aim (more adapt to the end of a paper introduction section). The authors should aknowledge eventual weaknesses of their research approch, discussing the way they would solve it in the next future.

Author Response

Response: Revised as Reviewer suggested at the end of Introduction and Result and Discussion.